# Target Therapy for Hepatocellular Carcinoma: Beyond Receptor Tyrosine Kinase Inhibitors and Immune Checkpoint Inhibitors

**DOI:** 10.3390/biology11040585

**Published:** 2022-04-12

**Authors:** Hyunjung Park, Hyerin Park, Jiyeon Baek, Hyuk Moon, Simon Weonsang Ro

**Affiliations:** Department of Genetics and Biotechnology, College of Life Sciences, Kyung Hee University, Yongin-si 17104, Korea; molly921@khu.ac.kr (H.P.); lealin@khu.ac.kr (H.P.); yeonii16@khu.ac.kr (J.B.); hmoon@khu.ac.kr (H.M.)

**Keywords:** hepatocellular carcinoma, target therapy, YAP/TAZ, Hedgehog, Wnt/β-catenin, animal models

## Abstract

**Simple Summary:**

Hepatocellular carcinoma (HCC) is the most common type of primary liver cancer and its incidence is steadily increasing. The development of HCC is a complex, multi-step process that is accompanied by alterations in multiple signaling cascades. Recent years have seen advancement in understanding molecular signaling pathways that play central roles in hepatocarcinogenesis. Aberrant activation of YAP/TAZ, Hedgehog, or Wnt/β-catenin signaling is frequently found in a subset of HCC patients. Targeting the signaling pathway via small molecule inhibitors could be a promising therapeutic option for the subset of patients. In this review, we will introduce the signaling pathways, discuss their roles in the development of HCC, and propose a therapeutic approach targeting the signaling pathways in the context of HCC.

**Abstract:**

Hepatocellular carcinoma (HCC) is a major health concern worldwide, and its incidence is increasing steadily. To date, receptor tyrosine kinases (RTKs) are the most favored molecular targets for the treatment of HCC, followed by immune checkpoint regulators such as PD-1, PD-L1, and CTLA-4. With less than desirable clinical outcomes from RTK inhibitors as well as immune checkpoint inhibitors (ICI) so far, novel molecular target therapies have been proposed for HCC. In this review, we will introduce diverse molecular signaling pathways that are aberrantly activated in HCC, focusing on YAP/TAZ, Hedgehog, and Wnt/β-catenin signaling pathways, and discuss potential therapeutic strategies targeting the signaling pathways in HCC.

## 1. Introduction

The incidence of liver cancer has been steadily increasing, posing a major global health problem. The World Health Organization reported that about 800,000 deaths were due to liver cancer, making it the fourth leading cause of cancer-related death [1]. Hepatocellular carcinoma (HCC) is the most common type of primary liver cancer, making up about 80% of the cases [2,3]. Surgical resection or local ablation therapy is performed for early-stage HCC, however, tumors recur in approximately 70% of these patients within 5 years [1,4]. Furthermore, although the diagnosis of HCC has improved significantly over past years, less than 30% of patients are diagnosed at the early stages when resection or local ablation is still available [1]. In the case of advanced HCC, systemic therapy is recommended as the standard treatment option, however, the prognosis has been unsatisfactory in general [1,2].

Molecular target therapy has been intensively studied in recent years as a promising treatment option for patients with advanced HCC [5,6,7]. The last decade has seen a significant advancement in molecular targeted therapy for HCC. Various small-molecule compounds targeting receptor tyrosine kinases (RTKs) have been tested in preclinical animal models and patients with HCC. So far, however, clinical outcomes of RTK-targeting molecular therapy have been less than satisfactory. For example, sorafenib is the leader compound among RTK inhibitors that are currently administered to patients with HCC, however, it has provided limited benefits for patients with HCC [7,8]. As a new therapeutic approach, immune checkpoint inhibitors have recently emerged a promising therapy for HCC [9,10,11,12,13]. Despite early excitements, targeting immune checkpoints alone has not shown prolonged therapeutic effects in HCC [8,9,13]. A combination of immune checkpoint inhibitors with other types of target therapy is currently considered most effective for the treatment of HCC and has been clinically tested.

Given the extreme heterogeneities in the molecular signature of HCC and difficulties in the development of globally-effective therapeutics for the disease, identification of molecular signaling pathways dysregulated in HCC would be fundamental to developing therapeutic targets as a personalized medical approach. In this review, we will introduce molecular signaling pathways beyond RTK that are aberrantly activated in HCC, that is, YAP/TAZ, Hedgehog, and Wnt/β-catenin signaling pathways, and discuss potential therapeutic strategies targeting the signaling pathways in the context of HCC.

## 2. Targeting YAP/TAZ in Liver Cancer

### 2.1. YAP/TAZ Signaling Pathway 

YAP/TAZ signaling is majorly activated when a tumor-suppressive Hippo signaling pathway is inactivated. A variety of mechanical signals such as cell shape and extracellular matrix (ECM) stiffness, as well as cell–cell interactions, elicit signals regulating the Hippo signaling cascade [14,15,16]. The Hippo signaling is also regulated by modulation of cell adhesion and cell polarity [17,18].

The Hippo–YAP/TAZ signaling pathway consists of mammalian sterile 20-like kinase 1 (MST1; also known as STK4) and MST2 (also known as STK3), large tumor suppressor kinase 1 (LATS1) and LATS2, the adaptor proteins Salvador 1 (SAV1), MOB1A and MOB1B, and yes-associated protein (YAP) and WW domain-containing transcription regulator protein 1 (WWTR1, also known as TAZ) [19,20] (Figure 1). The signal from the receptor goes through neurofibromatosis 2 (NF2) to MST1/2 [6,19,21]. When MST1/2 is activated by NF2, MST1/2 forms a complex with SAV1 and phosphorylate LATS1/2 and MOB1. Phosphorylated LATS1/2 then leads to phosphorylation of YAP and TAZ (YAP/TAZ). Phosphorylated YAP/TAZ bind to 14-3-3, resulting in cytoplasmic sequestration of the transcription factors, or are ubiquitylated, leading to protein degradation via the ubiquitin–proteasome pathway [19,22]. In contrast, when the Hippo signaling is turned off, upstream components of the Hippo signaling pathway become unphosphorylated, and this allows YAP/TAZ to translocate into the nucleus and activate the transcription of their target genes through the interaction with the TEA domain family members (TEAD1–TEAD4) [21,22]. Hippo-YAP/TAZ signaling has a major role in the regulation of cell proliferation, apoptosis, migration and differentiation, all essential for both developmental processes and homeostasis in adult organs [6,23,24,25,26].

### 2.2. YAP/TAZ Signaling in Liver Cancer

YAP/TAZ signaling is involved in multiple facets of carcinogenesis, including the promotion of cellular proliferation, maintenance of cancer stem cells (CSCs), and induction of tissue invasion of tumor cells, as well as drug resistance and recurrence of cancer [16,27,28,29]. In human HCCs, approximately 60% showed elevated levels of YAP/TAZ expression [30,31,32]. Of note, patients with liver cancer showing YAP overexpression have significantly low survival rates and high tumor recurrence rates, compared with those showing low levels of YAP expression [30,33].

Human HCC cells with high YAP or TAZ levels exhibit poor differentiation [28,34]. Moreover, when knockdown of TAZ was performed in human HCC cell lines, they revealed substantial declines in cell proliferation, migration, and invasion capabilities [35,36]. In addition, epithelial-mesenchymal transition (EMT) marker genes such as N-cadherin, vimentin, and snail, were downregulated following the knockdown of TAZ, indicating that TAZ promotes EMT of HCC cells [35,36]. In line with TAZ knockdown, when YAP was knocked down in human HCC cell lines, migration and invasion were significantly diminished [37,38]. Cell lines with a high YAP/TAZ level showed high capabilities of metastasis, as well [36]. Overall, the reports suggest that YAP/TAZ can contribute to multiple facets of HCC pathogenesis.

In addition to the roles played by YAP/TAZ in cell proliferation, apoptosis and migration, YAP/TAZ are also involved in hepatic fibrosis. Considering significant correlations between hepatic fibrosis and liver cancer, it is noteworthy that YAP/TAZ could initiate and promote hepatocarcinogenesis via upregulation of stromal activation and hepatic fibrosis [24,26]. Finally, YAP/TAZ play important roles in regulating the tumor microenvironment, cell metabolism, etc. [24,26,39,40].

### 2.3. Animal Models of Liver Cancer Induced by Activated YAP/TAZ Signaling

The precise roles of the Hippo–YAP/TAZ signaling pathway in hepatocarcinogenesis have been studied using genetically engineered mouse models in which various components of the signaling cascade were deleted or activated (Table 1). The studies consistently show that the inactivation of the Hippo signaling pathway increased YAP/TAZ activities, leading to liver overgrowth and ultimately the development of liver cancer [26,41,42].

Liver-specific deletion of NF2 using NF2^flox/flox^ mice and AAV-CRE (that is, CRE-expressing adeno-associated virus) led to the elevation of YAP/TAZ levels in the liver [45]. In NF2 knockout models, the proliferation of cells in bile duct areas and overgrowth of the liver were frequently observed, and these mice eventually developed HCC and cholangiocarcinoma (CCA) [43,46]. Of note, when YAP was additionally deleted in the NF2 knockout model, overgrowth of the liver and HCC incidence were reduced, indicating that hepatocarcinogenesis in the NF2 knockout models is mediated by activation of YAP.

In liver-specific MST1/2 knockout models, liver mass was generally increased due to elevated cell proliferation and ultimately HCC was induced. In the knockout models, proliferation and cell death were both increased, nevertheless, hepatomegaly is detected because the cell proliferation rate is much faster than the cell death rate [47,49]. In addition, nuclear accumulation of YAP increased in liver cells of the MST1/2 knockout mice via suppressing phosphorylation of YAP and Mob1 [19,48,51]. It should be noted that when either MST1 or MST2 was singly deleted, the liver size was not affected, suggesting that the MST1 and MST2 act in a redundant manner to control liver size [44,49,51]. In Mst1/2 knockout models, increases in the inflammatory reaction were also detected along with the upregulation of monocyte chemoattractant protein-1 (Mcp1), an inflammatory cytokine [50,62]. Genetic ablation of Mcp1 together with MST1/2 led to a reduction in inflammation, and ultimately to reduced tumor growth. In addition, when heterozygous deletion of YAP was induced in liver-specific MST1/2 knockout models, Mcp1 expression was inhibited and liver sizes returned to normal [50]. This strongly suggests that the inhibition of Hippo signaling induces liver inflammation via YAP-mediated transcriptional activation of Mcp1. Overall, MST1/2 determines the size of the liver by regulating the activity of YAP, and the suppression of MST1/2 can induce pro-tumorigenic inflammation in the liver and eventually HCC. Sav1 knockout models also showed hepatomegaly and the development of liver cancer, as seen in knockouts of other components of the Hippo signaling pathway [49,52]. In addition, double knockouts of Sav1 and PTEN dramatically accelerated tumorigenesis in the liver, compared with a single knockout of Sav1 [53].

Knockouts of Lats1/2 are mainly related to excessive biliary cell proliferation and carcinogenesis in the liver [55]. When the elevated YAP/TAZ activity was reduced by heterozygous deletion of YAP, hepatomegaly and hyperproliferation of bile duct cells were abolished [54,56]. These results show that Lats1/2 can control the liver size and biliary cell proliferation by acting as a negative upstream regulator of YAP. The dual deletion of Mob1A and Mob1B phenocopied characteristics of NF2 or Mst1/2 knockout models such as hepatomegaly, increased proliferation of bile duct cells, and incidences of HCC and CCA [57]. 

When YAP/TAZ was overexpressed in the liver via a transgenic approach, cellular proliferation was activated in the organ leading to increased liver volume without a visible tumor, whereas the knockout of YAP or TAZ led to decreased cell proliferation [63,64,65,66]. Recently, a simple liver-specific transgenic method has been developed via hydrodynamic tail vein injection (HTVI) [67,68,69,70,71]. The HTVI transgenesis allows an oncogene or oncogenic combinations to be expressed in hepatocytes. The expression of YAP or TAZ alone in the adult livers using the HTVI methodology did not lead to the development of tumors in the liver, but induced hyperproliferation and dedifferentiation of hepatocytes, suggesting requirements of other oncogenic partners for hepatocarcinogenesis in adults [67,72]. The co-expression of RAS together with TAZ led to the development of HCC at about 6 weeks after HTVI, while the expression of YAP plus PI3KCA induced CC-like tumors [60,61]. 

### 2.4. Preclinical Studies Targeting YAP/TAZ Signaling in HCC 

Small molecules targeting YAP/TAZ signaling are currently limited. Verteporfin (VP) is a small molecule inhibitor of YAP/TAZ that disrupts the interaction between YAP/TAZ and TEADs, suppressing transcriptional activities of YAP/TAZ [73,74]. It is also known to increase the level of cytosolic YAP/TAZ [75]. VP suppressed tumor cell proliferation in HCC as well as other types of cancers such as retinoblastoma [73], and colon cancer [76,77]. VP showed enhancement in anti-cancer effects when used in combination with other drugs. Sorafenib (SF) is a multi-kinase inhibitor that is currently widely used for patients with HCC. Although SF alone showed minimal anti-cancer effects, treatment with VP and SF together exerted a synergistic effect on tumor cell death. Xenograft HCC models also showed similar results [78]. Using a subcutaneous xenograft tumor model, the combination of VP and other anti-cancer drugs such as 5-FU or Doxorubicin led to significantly reduced tumor growth by elevating apoptosis rates of tumor cells [79].

Vestigial Like Family Member 4 (VGLL4) is a tumor suppressor that competes with YAP in binding to TEADs, and thus, inhibits the transcriptional activity of YAP [16,76]. Finding that domains in the carboxyl-terminal of VGLL4 (TDU domain) are sufficient for the interaction with TEADs, short peptides mimicking the TDU domain have been developed, among which Super TDU is noteworthy [27,80,81]. In gastric cancer, treatment with Super-TDU effectively suppressed the proliferation of cancer cells. Further, Super-TDU suppressed HCC in subcutaneous and intrahepatic xenograft models via the inhibition of YAP/TAZ activities [82].

Sitagliptin is an oral hypoglycemic agent that is widely used in many countries to treat diabetes [83]. Its mechanism of action is through the inhibition of dipeptidyl peptidase-4 (DPP-4), an enzyme that degrades and inactivates glucagon-like peptide-1 (GLP-1). The elevated GLP-1 level in response to sitagliptin leads to increased insulin release and improves glucose tolerance. Recently, it was found that sitagliptin inhibited YAP nuclear translocation [83]. The inhibitory effect on YAP exerted by sitagliptin is mediated via its activation of AMP-activated protein kinase (AMPK), which does not only promote phosphorylation of LATS1/2 but also directly phosphorylates YAP, leading to cytoplasmic retention and degradation of the transcription factor [83]. Besides sitagliptin, other AMPK agonists such as metformin and phenformin can also exert similar inhibitory effects on YAP. In chemically induced HCC mouse models, the treatment of tumor-bearing mice with sitagliptin at a dose of 80 mg/kg for 60 days significantly prolonged the survival of the treated mice and reduced malignancies of HCC in the animals [84].

## 3. Targeting Hedgehog Signaling in Liver Cancer

### 3.1. Hedgehog Signaling Pathway

The Hedgehog genes were first identified in the late 1970s through the genetic screening of mutations leasing to “spiked” phenotypes of the cuticle of Drosophila [85,86,87,88]. It has been later found that the genes are highly conserved from Drosophila to mammals. Hedgehog signaling regulates cell proliferation, differentiation, tissue homeostasis, and carcinogenesis [85,87,89,90,91]. In humans, there are three types of Hedgehog ligands: Sonic Hedgehog (Shh), Indian Hedgehog (Ihh) and Desert Hedgehog (Dhh) [86,87]. 

The canonical Hedgehog signaling pathway is activated when Hedgehog ligands bind to the receptors, Patched (Ptch) proteins (Figure 2) [87]. In the absence of Hedgehog ligands, Ptch continues to repress the activity of Smoothened (Smo), a G-protein-coupled receptor-like protein and inhibits translocation of Smo to the primary cilium (PC) [86,87,89,90,92,93]. When Smo is not located in PC, a Glioma-associated oncogene (Gli) is phosphorylated by protein kinase A (PKA), glycogen synthase kinase-3 (GSK3), and casein kinase 1 (CK1) [89,90,94,95,96,97]. Phosphorylated Gli undergoes proteolytic cleavage by β-transducin-repeat-containing protein (β-TrCP), generating Gli repressor (GliR) [89,94,95,96,98]. GliR binds to the promoters of Hedgehog target genes, suppressing their transcription. In contrast, the binding of Hedgehog ligands to Ptch relieves the suppression of Smo by the receptor protein, leading to the translocation of Smo to PC [87,92,99]. The presence of Smo in PC suppresses phosphorylation and proteolytic cleavage of Gli. It is also suggested that Smo activates Gli by relieving inhibition of Gli by kinesin protein (Kif7) and Suppressor of fused (Sufu) [89,95,98]. The release of Gli from the inhibitory protein complex allows Gli to be active (GliA) and to translocate into the nucleus where it activates the transcription of a plethora of Hedgehog target genes [87,89]. In addition to the canonical Hedgehog pathway, there are two types of non-canonical Hedgehog pathways that do not involve Gli-mediated transcription regulation of hedgehog target genes [94,100]. Type I non-canonical pathway is Smo-independent, while Type II is Smo-dependent. Non-canonical Hedgehog signaling pathways are beyond the scope of this review and detailed information can be found elsewhere [94,99,100,101,102].

### 3.2. Hedgehog Signaling in Liver Cancer

The synthesis of hedgehog ligands is stimulated by diverse factors that trigger liver regeneration, which include various liver mitogens and cell stressors. The Hedgehog ligands are released from ligand-producing cells into the local environment where they engage receptors on Hedgehog responsive cells. In particular, Sonic Hedgehog (Shh), the representative Hedgehog ligand, enhances the viabilities of Hedgehog target cells, serving as viability and proliferative factors for various types of liver cells [89,94]. Shh is overexpressed in approximately 60% of human HCC, suggesting that Hedgehog pathway activation contributes to hepatocarcinogenesis [103,104,105,106].

Shh is strongly expressed in the ventral foregut endoderm where liver, pancreas and lung buds develop [94]. Shh is also expressed temporarily in hepatoblasts later in development [94]. As hepatoblasts differentiate into hepatocytes, the expression of Shh decreases [94]. Therefore, the Shh pathway is activated in liver embryogenesis, but it is mostly inactive in mature hepatocytes in healthy adult liver [107]. The Shh pathway can be re-activated in adult livers when acute or chronic liver regeneration is induced. Various liver cells such as hepatocytes, cholangiocytes, myofibroblastic stellate cells, sinusoidal endothelial cells, and immune cells produce Shh for wound healing in an injured liver [89,94]. 

The elevated expression of Shh and its target genes such as Ptch and Gli was detected in HCC tissues [104,107,108,109]. It is also demonstrated that the Shh signaling pathway elicits cell migration and invasion of HCC cells. Down-regulation of Gli using siRNA inhibited adhesion, migration and invasion of HCC [110]. Further, treatment with Shh enhanced migration and invasion of HCC cells such as LO2, SMMC-7721, and SK-Hep [111,112]. 

### 3.3. Animal Models of Liver Cancer Induced by Activated Hedgehog Signaling

Various studies employing animal models have demonstrated that the Hedgehog signaling pathway plays an important role in hepatocarcinogenesis. A transgenic mouse model was developed in which Shh is secreted from hepatocytes, which mimics histopathological features of patients with non-alcoholic fatty liver disease (NAFLD) and non-alcoholic steatohepatitis (NASH) [113]. Secreted Shh in the model activated Hedgehog signaling in numerous cells of various types in the hepatic tissue and led to hepatic fibrosis and hepatocarcinogenesis [114]. Moreover, the Mdr2 knockout model consistently expressed Hedgehog ligands and progressively accumulated myofibroblasts and progenitors in the liver with age [105]. Activated Hedgehog signaling in the model led to chronic inflammation, liver injury and fibrosis, and finally HCC [105]. Of note, the treatment of Mdr2-deficient mice with an inhibitor of Hedgehog signaling suppressed liver fibrosis and HCC [105]. 

Genetic ablation of Smo in myofibroblasts can be achieved using SMO^flox/flox^; αSMA-Cre^ERT2^ mice in which treatment with tamoxifen induces nuclear import of Cre recombinases in myofibroblasts and leaves out an exon of Smo gene [115]. The treatment of SMO^flox/flox^; αSMA-Cre^ERT2^ mice with tamoxifen showed a blockade of Hedgehog signaling selectively in myofibroblasts and inhibited liver fibrosis [115]. This suggests that the activation of Hedgehog signaling in myofibroblasts is critical in inducing liver fibrosis. Considering the strong correlation between hepatic fibrosis and liver cancer, the data suggest that Hedgehog activation in hepatic stroma promotes liver cancer.

### 3.4. Preclinical Studies Targeting Hedgehog Signaling in HCC

Preclinical studies targeting the Hedgehog signaling pathway mainly focus on the inhibition of the activity of Smo. For example, cyclopamine can block the Hedgehog signaling pathway by inhibiting Smo. It is demonstrated that cyclopamine increased apoptosis and decreased cells growth and proliferation in various HCC cell lines including SMMC-7721, SK-Hep1, Hep3B, Huh7, and PLC [103,104,108,116]. Vismodegib is also a potent inhibitor of Smo. In in vivo preclinical studies, both cyclopamine and vismodegib inhibited the growth of tumors in the liver [117]. Hepatitis B virus X protein-expressing transgenic mice (HBxTg) develop hepatitis and steatosis by 5 to 6 months of age, dysplastic nodules by 8 to 9 months, and visible HCC by 12 months [118]. Of note, the HBxTg mice exhibited increased levels of Gli2 and Shh in the livers. Treatment of the HBxTg mice with vismodegib downregulated the levels of Gli2 and Shh and more importantly, suppressed tumor development in their livers [118].

Taccalonolide A is a Hedgehog antagonist. Treatment with Taccalonolide A decreased mRNA levels of Hedgehog signaling molecules and led to reduced viability of HCC cells [119]. GANT61 is a selective inhibitor of Gli-mediated transactivation [120,121]. Preclinical studies were performed for numerous cancer types, including rhabdomyosarcoma, neuroblastoma, and leukemia, as well as colon and pancreatic cancer. In HCC, treatment with GANT61 significantly reduced cell proliferation and viability [120]. In an HCC xenograft model transplanted with Huh7 cells, the administration of GANT61 inhibited tumor formation and growth [120].

## 4. Targeting Wnt/β-Catenin Signaling in Liver Cancer

### 4.1. Wnt/β-Catenin Signaling Pathway

The Wnt/β-catenin signaling is a signal transduction pathway that initiates with the binding of Wnt protein to its receptor Frizzled and then transmits the signal through β-catenin. In the absence of Wnt, β-catenin is downregulated by a destruction complex which consists of Glycogenesis kinase3β (GSK3β) and casein kinase1α (CK1α), tumor suppressor adenomatous polyposis coli (APC), and scaffold protein AXIN (Figure 3) [122,123]. The β-catenin destruction complex phosphorylates the N-terminus of β-catenin through GSK3β and CK1α [124,125]. Phosphorylated β-catenin is ubiquitinated by β-transducin-repeat-containing protein (β-TrCP) ubiquitin ligase and destroyed by the proteasome [124,126]. Endogenous Wnt antagonists such as secreted frozen-related proteins (SFRPs), Dickkopfs (DKKs), and Wnt Inhibitory factors (WIFs) suppress Wnt/ β-catenin signaling by inhibiting the interaction between Wnt ligands and the receptors, Frizzled [124,127,128,129,130]. 

The binding of Wnt to receptors Frizzled on the cell surface recruits phosphoprotein Dishevelled (Dvl) to the receptor, providing a binding site for Axin and GSK3β. This event leads to phosphorylation of transmembrane low-density lipoprotein receptor-related protein5/6 (LRP5/6), and more importantly inhibits phosphorylation of β-catenin by GSK3β [6,131]. Unphosphorylated β-catenin can escape from ubiquitin-mediated protein degradation, and thus, increases its concentration in the cytoplasm [6,131]. Accumulated β-catenin can translocate into the nucleus, where it interacts with the T-cell factor/lymphoid enhancer-binding factor (TCF/LEF) for transcriptional activation of numerous target genes [122,132]. Genes related to cell proliferation (MYC, MYB, CYD1, etc.), angiogenesis (VEGF and c-MET), and anti-apoptosis (Bcl-2 and Survivin) are some of the target genes of Wnt/β-catenin signaling [6,133,134,135]. There are other transcription factors such as hypoxia-inducible factor 1α (HIFIα), forkhead box protein O (FOXO), and several members of the sex-determining region Y box (SOX) family with which β-catenin associates for transcriptional activation [136,137,138,139,140].

### 4.2. Wnt/β-Catenin Signaling in Liver Cancer 

Studies have shown that there is a strong correlation between the level of nuclear β-catenin and the tumor grade of HCC [141,142]. Elevated levels of nuclear β-catenin in HCC have also been associated with increased proliferation of tumor cells in HCC [143,144]. Aberrant activation of the Wnt/β-catenin signaling pathway or deregulated nuclear accumulation of β-catenin can be caused by mutations in components of the pathways, which are frequently found in HCC [136,137]. Accumulation of β-catenin plays important role in HCC development. About 20–35% of HCCs are caused by genetic mutations in the *CTNNB1* gene (coding for β-catenin), rendering the protein resistant to degradation [136,145]. Three-fourths of *CTNNB1* mutations are caused by missense substitution which occurs at exon 3 at amino acid position 29–49 [136,146,147]. The exon 3 of β-catenin has multiple target sites for phosphorylation by GSK3 as well as ubiquitination sites for degradation by the proteasome [148,149]. Thus, mutation at these phosphorylation and ubiquitination sites can enhance the stability of β-catenin. 

The fact that the accumulation of β-catenin in HCC is more frequently found than mutations in *CTNNB1* strongly suggests that loss-of-function mutation in negative regulators of β-catenin such as APC or AXIN should be present in HCC. In line with the speculation, a loss-of-function mutation in *AXIN* occurs at a frequency of 3–16% in HCC, which is the second most common mutation in HCC that leads to activation of the Wnt/β-catenin signaling pathway [124,146,150]. The AXIN mutation suppresses the degradation of β-catenin by inhibiting its phosphorylation [124,151,152]. Satoh et al. reported that they found 13 genetic alterations in *CTNNB1* among 100 primary HCC tissues. Of note, they found 6 mutations in *AXIN1* in the 87 tumor samples that had no *CTNNB1* mutations, underlining the significance of *AXIN1* mutation in human HCC [124,151,152]. Unlike mutations in *CTNNB1*, 37.5% of mutations found in *AXIN* are nonsense mutations, while only 30% of mutations are attributed to missense mutations [133,136]. 

### 4.3. Animal Models of Liver Cancer Induced by Activated Wnt/β-Catenin Signaling 

Although aberrant activation of β-catenin is frequently found in human HCC, the expression of a constitutively active form of β-catenin alone was not enough to develop HCC in murine models [153,154]. This strongly suggests that the Wnt/β-catenin signaling pathway requires an oncogenic collaborator for the development of HCC [155,156,157]. Genetically engineered mouse models have revealed that activated Wnt/β-catenin signaling can effectively cooperate with activated RAS, MET, and AKT signaling pathways to induce HCC (Table 2). For example, the expression of a constitutively active form of NRAS (NRAS^G12V^) alone did not induce HCC, while the co-expression of NRAS^G12V^ with an activated mutant form of β-catenin led to the formation of HCC [157]. The simultaneous expression of c-Met and active β-catenin also developed HCC at 12 weeks following the expression [158]. Considering that one-fifth of human HCC showed overexpression/activation of both c-Met and β-catenin, the double transgenic mouse models well represent human HCC of this subset [158,159].

The co-activation of both the Wnt/β-catenin and Akt/mTOR pathways is found in about 10% of patients with HCC who frequently show poor survival. Toh et al. reported that the co-expression of constitutively active forms of human AKT and β-catenin led to the formation of a high HCC tumor burden in both male and female FVB/N mice with 100% penetrance [160]. Of note, tumors induced by the co-activation of AKT and Wnt/ β-catenin signaling pathways contained a subpopulation of cells with stem/progenitor-like characteristics, which may contribute to tumor self-renewal and drug resistance [160].

### 4.4. Preclinical Studies Targeting Wnt/β-Catenin Signaling in HCC

Based on the frequent activation of Wnt/β-catenin signaling in HCC, various molecules targeting the signaling pathway have been tested for HCC treatment. In an attempt to block the initiation step of the signaling cascade, monoclonal antibodies such as OMP-18R5 have been developed which bind to Frizzled receptors and inhibit its interaction with Wnt ligands. OMP-18R5 showed suppression of tumors in xenograft models of various types of cancers [161]. OMP-54F28 is a fusion protein combining the frizzled family receptor 8 (FZD8) with the immunoglobulin Fc domain which is designed to compete with the native Frizzled receptors for Wnt ligands. OMP-54F28 suppressed Wnt activation and reduced tumor growth in patient-derived xenograft models [162,163]. 

Considering that Frizzled receptor 7 (FZD7) is most frequently overexpressed in HCC among the 10 human Frizzled receptors [164], small interfering peptides were developed that disrupt the interaction of FZD7 with Dvl (Figure 3). The treatment of HCC cells with the peptides containing the Dvl-binding motif of FZD7 decreased the cell viability of the tumor cells through β-catenin degradation [165]. Fz7-21 is another peptide antagonist of Frizzled receptor 7 which is isolated from a peptide phage library [166]. The peptide specifically binds to FZD7 and impairs the interaction of FZD7 with Wnt ligands. Two human hepatoma cell lines, HepG2 and Huh-7 with high expression levels of FZD7 were selected to test the efficacy of Fz7-21 as an antagonist of Wnt/β-catenin signaling. Treatment with Fz7-21 strongly inhibited the Wnt/β-catenin signaling pathway in the cell lines and subsequently induced apoptosis and anti-proliferative effects on HCC cells [167]. 

CGP049090, PKF118-310, and PKF115-584 are small molecules that inhibit the interaction between Tcf4 and β-catenin, and thus suppress the transcriptional activity of the Tcf4/β-catenin complex. In in vitro studies, treatment with the inhibitors induced apoptosis and cell cycle arrest in human hepatoma cell lines, HepG2, Hep40, and Huh-7 [168,169]. Further, in athymic nude mice transplanted with HepG2 cells, treatment with each of the chemicals significantly suppressed the growths of tumors in the xenograft models. Molecular analysis of harvested tumor tissues revealed substantially increased apoptosis in tumors treated with the inhibitors. Further, tumors treated with the inhibitors showed reduced expression levels of c-Myc, cyclin D1 and survivin, which are transcriptional targets of TCF4/β-catenin [168,170]. 

Tankyrase acts as an activator of Wnt/β-catenin signaling by mediating the degradation of AXIN1 and AXIN2 which are negative regulators of β-catenin [170]. NVP-TNKS656 inhibits tankyrase by binding to the dual active pockets in tankyrase [171]. In HCC cell lines such as SMMC-7721 and MHHC97H, NVP-TNKS656 blocked cell proliferation and clone formation [172]. Further, the drug also suppressed migration and invasion of HCC cells [171,172,173]. Anther tankyrase inhibitor, XAV939, also inhibited cell proliferation and colony formation in HCC cell lines (Table 3). In xenograft models transplanted with HepG2, intra-tumor injections of XAV939 significantly suppressed tumor growth [174].

## 5. Perspectives and Conclusions

The development of HCC is a complex, multi-step process accompanied by alterations in multiple signaling cascades. A fundamental understanding of molecular signaling pathways to tumorigenesis can help predict patient response to targeted therapies, which can have a substantial impact on clinical decision-making. Of all the tumor driver genes, those responsible for oncogenic addiction are important [175,176]. Accumulating data have shown that YAP/TAZ, Hedgehog, and Wnt/β-catenin signaling pathways play central roles in carcinogenesis of HCC and could be the promising Achilles’ heel to attack cancer.

Most studies targeting the YAP/TAZ, Hedgehog, or Wnt/β-catenin signaling pathways in HCC are currently under preclinical stages in which the target therapies have reduced cell viability, proliferation and migratory potentials of HCC. Phase I and/or II clinical trials targeting the signaling pathways have been completed or are undergoing (Table 4). With the advancement in drug discovery and screening, safe and effective target therapeutics will be developed and tested for HCC in clinical settings. For instance, BC2059 and DKN-01 are drugs targeting the Wnt//β-catenin signaling pathway, and they have shown tumor suppression in murine models of acute myeloid leukemia and ovarian cancer, respectively [177,178]. The target therapeutics wait for phase I/II clinical trials for patients with HCC. We look forward to seeing how effective the novel target therapies will be in HCC in clinical trials.

## Author Contributions

All authors have read and agreed to the published version of the manuscript.

## Figures and Tables

**Figure 1 biology-11-00585-f001:**
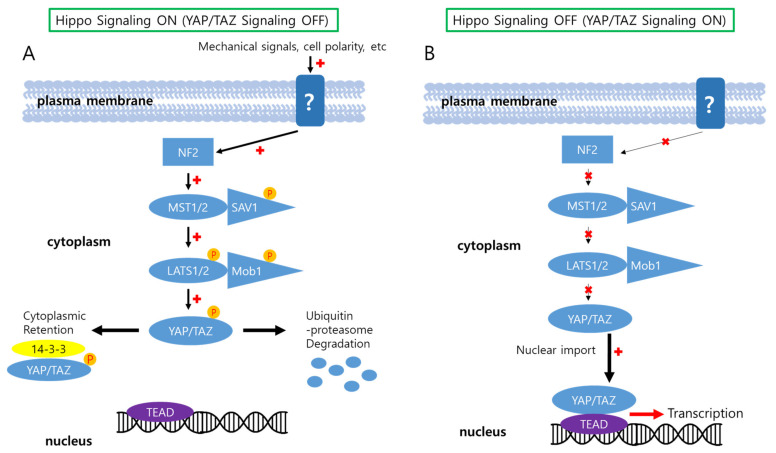
Schematic illustration of Hippo-YAP/TAZ signaling pathway. A variety of cellular conditions in tissue such as cell density, extracellular matrix (ECM) stiffness, or cell polarity can elicit signals regulating the Hippo signaling cascade. Receptors sensing the cellular conditions have yet to be identified. (**A**) When the Hippo signaling is turned on, sequential phosphorylation processes by kinase components of the pathway lead to phosphorylation of YAP/TAZ, resulting in cytoplasmic retention or degradation of the transcription factors. (**B**) When the Hippo signaling is turned off, unphosphorylated YAP/TAZ can translocate into the nucleus, where they interact with TEAD and transcriptionally activate numerous target genes. The Hippo–YAP/TAZ signaling pathway presented in the figure is tightly regulated in hepatocytes, the parenchymal cells in the liver.

**Figure 2 biology-11-00585-f002:**
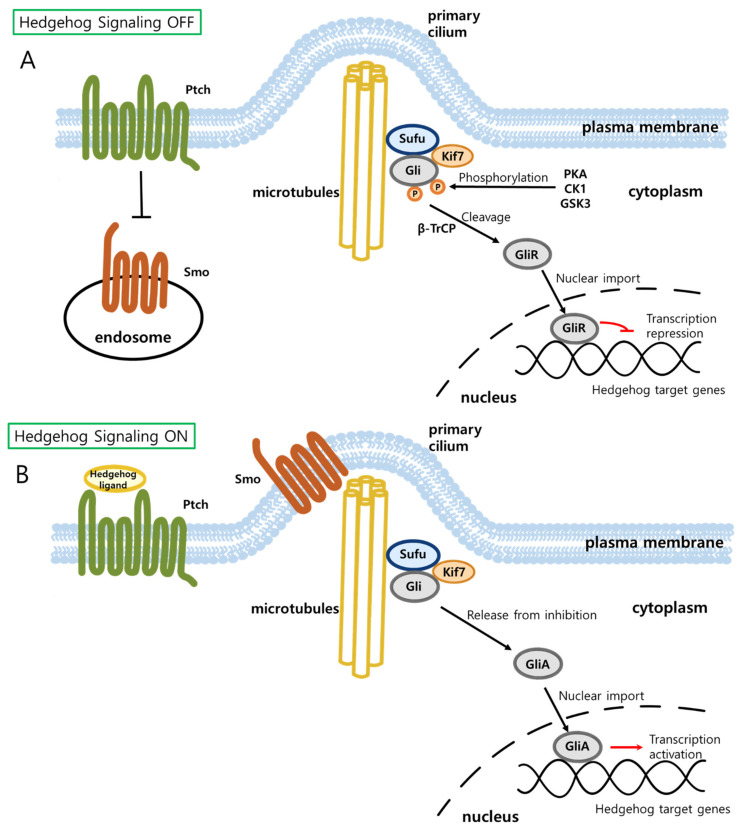
Schematic illustration of signal transduction by the canonical Hedgehog signaling pathway. (**A**) In the absence of the ligand, the patched receptor (Ptch) suppresses Smo, leading to the generation of Gli repressor (GliR) by proteolytic cleavage of Gli. GliR suppresses transcription of Hedgehog target genes. (**B**) Binding of ligands to Ptch relieves the suppression of Smo by the receptor, leading to movement of Smo to primary cilium where it inhibits proteolysis of Gli. Uncleaved and unphosphorylated Gli protein translocates into the nucleus and induces transcription of Hedgehog target genes as Gli transcriptional activator (GliA). Ptch, Patched; Smo, Smoothened; Gli, Glioma-associated oncogene.

**Figure 3 biology-11-00585-f003:**
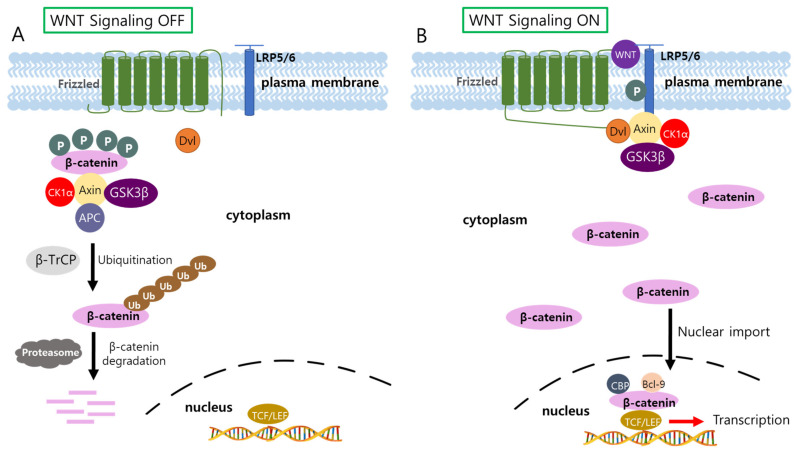
Schematic illustration of signal transduction by Wnt/ β-catenin signaling pathway. (**A**) In the absence of the ligand Wnt, β-catenin is degraded via the ubiquitin-mediated proteasome pathway. (**B**) In the presence of the ligand, β-catenin escapes from β-catenin destruction complex and translocates into the nucleus where it induces transcriptional activation of a plethora of target genes through the interaction with TCF/LEF.

**Table 1 biology-11-00585-t001:** Genetically engineered mouse models with dysregulated Hippo-YAP/TAZ signaling.

Gene	Mouse Model	Phenotype/Tumor Type	Reference
NF2	Alb-Cre; NF2 f/f	Bile duct hamartomas, HCC	[43,44]
	AVV-Cre; NF2 f/f	Increased ductular structure	[45]
	Alb-Cre; NF2 f/f	Hepatomegaly, HCC, CCA	[46]
MST1/2	Alb-Cre; Mst1/2 f/f	Hepatomegaly, Dysplasia, Increased cell death	[47]
	Alb-Cre; Mst1/2 f/f	Liver overgrowth, Increased cell proliferation	[48]
	Alb-Cre; Mst1/2 f/f	Hepatomegaly, HCC, CCA	[49]
	Alb-Cre; Mst1-/-; Mst2 f/f	Increased pro-inflammatory cytokines	[50]
	Ad-Cre; Mst1-/-, Mst2 f/-	Hepatomegaly, HCC	[51]
Sav1	Alb-Cre; Sav1 f/f	Hepatomegaly, Increased immature progenitor cells	[52]
	Alb-Cre; Sav1 f/f	Increased cell proliferation, HCC, CCA	[49]
	Alb-Cre; Sav1 f/f	Hepatomegaly, HCC, CCA	[53]
LATS1/2	Alb-Cre; Lats1 f/f; Lats2 f/f	Bile duct malformation, High lethality	[54]
	Alb-Cre; Lats1-/-; Lats2 f/f	High immature BECs proliferation	[55]
	Alb-Cre; Lats1-/-; Lats2 f/f	Hepatomegaly, BECs Proliferation	[56]
MOB1	Alb-Cre; Mob1a f/f; Mob1b-/-	Increased cholangiocyte-like cells and oval cells, HCC	[57]
YAP/TAZ	ApoE-rtTA; TRE-hYAP	Hepatomegaly, Increased cell proliferation	[58]
	pCMV-Cre; Yap f/f, Taz f/f	Reduced tumor cell proliferation	[34]
	Alb-Cre; Mst1/2 f/f; Yap+/-	Necrosis, Cholestasis, Fibrosis, Swelling of tissue	[48]
	LAP1-tTA; TetO-YAPS127A	Hepatomegaly, Nuclei enlargement	[59]
	AAV-Cre; TetO-YAPS127A	Rapid liver growth	[45]
	TAZ^S89A^ + HRAS^G12V^	Tumor stromal activation, HCC	[60]
	YAP^S127A^ + PIK3CA^H1047R^	High lipid hepatocytes, HCC, CCA	[61]

**Table 2 biology-11-00585-t002:** Development of HCC in murine models expressing destruction-resistant β-catenin and its oncogenic partner.

Genes	Latency	Tumor Type	Reference
CMet + ΔN90-β-catenin	~12 weeks	HCC	[158]
NRAS^G12V^ + ΔN90-β-catenin	~12 weeks	HCC	[157]
Spry2Y55F + ΔN90-β-catenin	~24 weeks	HCC	[157]
myr-AKT + ΔN90-β-catenin	~13 weeks	HCC	[160]

**Table 3 biology-11-00585-t003:** Preclinical studies targeting Wnt/β-catenin signaling in HCC.

Drug	Target	Phase	Cell Line	Mouse Model	Reference
Fz7-21	FZD7	In vitro	HepG2, Huh-7	Not determined	[166]
CGP049090	β-catenin/TCF	In vivo	HepG2, Hep40, Huh-7	Xenograft	[168,169]
PFK118-310	β-catenin/TCF	In vivo	HepG2, Hep40, Huh-7	Xenograft	[168,169]
PFK115-584	β-catenin/TCF	In vivo	HepG2, Hep40, Huh-7	Xenograft	[168,169]
NVP-TNKS656	Tankyrase	In vitro	SMMC-7721, MHHC-97h	Not determined	[171]
XAV939	Tankyrase	In vivo	HepG2, Hep40, Huh-7	Xenograft	[174]

**Table 4 biology-11-00585-t004:** Currently undergoing clinical trials targeting YAP/TAZ, Hedgehog Wnt/β-catenin signaling in HCC.

Drug	Target Signaling	NCT Number	Phase	Current Status	Reference
Sitagliptin	YAP/TAZ	NCT02650427	I	Completed	[179]
Sonidegib	Hedgehog	NCT02151864	I	Completed	[180]
Vismodegib	Hedgehog	NCT01546519	I	Completed	[181]
OMP-54F28	Wnt/β-catenin	NCT02069145	I	Completed	[182]
DKN-01	Wnt/β-catenin	NCT03645980	I/II	Recruiting	[183]
BC2059	Wnt/β-catenin	NCT04851119	I/II	Recruiting	[184]

## Data Availability

No new data were created or analyzed in this study. Data sharing is not applicable to this article.

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
