# Peer review of "Target Therapy for Hepatocellular Carcinoma: Beyond Receptor Tyrosine Kinase Inhibitors and Immune Checkpoint Inhibitors"

_biology, 2022, doi:10.3390/biology11040585_

Round 1

Reviewer 1 Report

Figure 1. Overall, this figure is so general without any detailed or important information. Labeling is also missing. Please add localization to the figure such as plasma membrane, cytoplasm, and nucleolus for each step of the pathways. Please clarify what type of cells are you presenting. The graph on the right doesn’t have any arrow for description of the pathway which makes it hard to understand. Please include activation or inhibition signs for the right graph.

Line 110: The sentence: Overall, the reports strongly support the fact that YAP/TAZ are critical modulators controlling the pathogenesis of HCC sounds overstated since all the discussed references have used cell lines and no animal study have mentioned. It would be more appropriate to smooth down this sentence.

Line 160: In this sentence: “When YAP/TAZ was overexpressed in the liver via transgenic approach, liver size increased rapidly, and cell proliferation was also activated, whereas knockout of YAP or TAZ led to decreased cell proliferation” are we talking about the size of normal liver after transfection or the size of the HCC tumor? Please clarify.

Line 194: Please provide some information about mechanisms of action for Sitagliptin.

Line 204: The pathway description sounds too complicated since the sentences don’t follow the same story. Can you please enhance the flow of the sentences in those 2 paragraphs?

Line 221: Is SMO the same as Smo (the G-protein coupled receptor, mentioned in line 213)? Please clarify.

Line 404: What does “to antagonize the binding of FZD7 to Dvl” mean?

Figure 2. Please clarify abbreviations such as PC, Hh, and Smo in the legend. Please label the tubular structure in yellow.

Figure 3. Please label plasma membrane, cytoplasm, and nucleolus. What are the arrows showing? Activation? Phosphorylation? Ubiquitination? Please clarify.   

Table 3, and 4. Please add the references to the tables.

Author Response

Figure 1. Overall, this figure is so general without any detailed or important information. Labeling is also missing. Please add localization to the figure such as plasma membrane, cytoplasm, and nucleolus for each step of the pathways. Please clarify what type of cells are you presenting. The graph on the right doesn’t have any arrow for description of the pathway which makes it hard to understand. Please include activation or inhibition signs for the right graph.

Response: In revisions, we made changes in Figure 1 and added more information to the figure, according to the reviewer’s suggestions. The type of cells presented in the figure is hepatocytes which we indicated in the legends in the revised manuscript. We are grateful for the important suggestions.

Line 110: The sentence: Overall, the reports strongly support the fact that YAP/TAZ are critical modulators controlling the pathogenesis of HCC sounds overstated since all the discussed references have used cell lines and no animal study have mentioned. It would be more appropriate to smooth down this sentence.

 Response: In revisions, we made a change in the sentence not to overstate it by saying “Overall, the reports suggest that YAP/TAZ can contribute to multiple facets of HCC pathogenesis”.

Line 160: In this sentence: “When YAP/TAZ was overexpressed in the liver via transgenic approach, liver size increased rapidly, and cell proliferation was also activated, whereas knockout of YAP or TAZ led to decreased cell proliferation” are we talking about the size of normal liver after transfection or the size of the HCC tumor? Please clarify.

 Response: We meant the size of normal liver. For clarification, we rephrased it as “ When YAP/TAZ was overexpressed in the liver via transgenic approach, cellular proliferation was activated in the organ leading to increased liver volume without a visible tumor, whereas knockout of YAP or TAZ led to decreased cell proliferation”.

Line 194: Please provide some information about mechanisms of action for Sitagliptin.

Response: We provided mechanisms of action for Sitagliptin in more detail in the revised manuscript.

Line 204: The pathway description sounds too complicated since the sentences don’t follow the same story. Can you please enhance the flow of the sentences in those 2 paragraphs?

Response: We agree that the descriptions in the paragraphs were unnecessarily expanded and caused complications. The non-canonical pathway of Hedgehog signaling is not within the scope of our manuscript, so we briefly mentioned it and left out the detailed descriptions in revisions. We appreciate the important suggestion.

Line 221: Is SMO the same as Smo (the G-protein coupled receptor, mentioned in line 213)? Please clarify.

Response: Yes, it is. We changed “SMO” to Smo in the revised manuscript.

Line 404: What does “to antagonize the binding of FZD7 to Dvl” mean?

 Response: We apologize the lack of clarity in the sentence. The sentence was changed to “….small interfering peptides were developed that disrupt the interaction of FZD7 with Dvl”.

Figure 2. Please clarify abbreviations such as PC, Hh, and Smo in the legend. Please label the tubular structure in yellow.

Response: In the revised manuscript, we labeled the tubular structure in yellow in the figure, which is microtubules. As well, we clarified the abbreviations in the legends. 

Figure 3. Please label plasma membrane, cytoplasm, and nucleolus. What are the arrows showing? Activation? Phosphorylation? Ubiquitination? Please clarify.   

Response: In revisions, we labeled the cellular components and the processes that the arrows indicate, as the reviewer suggested.

Table 3, and 4. Please add the references to the tables.

Response: We added references to the tables in the revised manuscript. We appreciate the suggestion.

Reviewer 2 Report

The review manuscript title "Target Therapy for Hepatocellular Carcinoma: Beyond Receptor Tyrosine Kinase Inhibitors and Immune Checkpoint inhibitors" was well written and covered comprehensively all information relevant to the topic. The review will be more beneficial to the young researcher and brings an overview of the current challenges in the HCC. I recommend the article for acceptance in the current format.

Author Response

The review manuscript title "Target Therapy for Hepatocellular Carcinoma: Beyond Receptor Tyrosine Kinase Inhibitors and Immune Checkpoint inhibitors" was well written and covered comprehensively all information relevant to the topic. The review will be more beneficial to the young researcher and brings an overview of the current challenges in the HCC. I recommend the article for acceptance in the current format.

Response: We sincerely appreciate the reviewer’s positive evaluation of the manuscript.